# Genome-Wide Identification and Analysis of DNA Methyltransferase and Demethylase Gene Families in Sweet Potato and Its Diploid Relative

**DOI:** 10.3390/plants14111735

**Published:** 2025-06-05

**Authors:** Songtao Yang, Shuai Qiao, Yan Yang, Fang Wang, Wei Song, Wenfang Tan, Yongping Li, Youlin Zhu

**Affiliations:** 1College of Life Science, Nanchang University, Nanchang 330031, China; yost@scsaas.cn (S.Y.); 13210779656@163.com (Y.Y.); 2Sichuan Germplasm Resources Center, Crop Research Institute, Sichuan Academy of Agricultural Sciences, Chengdu 610066, China; qiaoshuai@scsaas.cn (S.Q.); wangfangsaas@126.com (F.W.); songweizws@163.com (W.S.); zwstwf@scsaas.cn (W.T.); 3Environmentally Friendly Crop Germplasm Innovation and Genetic Improvement Key Laboratory of Sichuan Province, Chengdu 610066, China; 4School of Breeding and Multiplication (Sanya Institute of Breeding and Multiplication), Hainan University, Sanya 572025, China; 5Key Laboratory for Quality Regulation of Tropical Horticultural Crops of Hainan Province, School of Tropical Agriculture and Forestry, Hainan University, Haikou 570228, China

**Keywords:** sweet potato, DNA methylation, C5-MTase, dMTase, storage root development

## Abstract

DNA methylation is a conserved and vital epigenetic modification that plays essential roles in plant growth, development, and responses to environmental stress. Cytosine-5 DNA methyltransferases (C5-MTases) and DNA demethylases (dMTases) are key regulators of DNA methylation dynamics. However, a comprehensive characterization of these gene families in sweet potato has remained elusive. In this study, we systematically identified and analyzed eight C5-MTase and five dMTase genes in the genomes of diploid (*Ipomoea trifida*, 2n = 2x = 30) and autohexaploid (*Ipomoea batatas*, 2n = 6x = 90) sweet potato. Phylogenetic, structural, and synteny analyses revealed a high degree of conservation among these genes, suggesting their essential roles during evolution. Promoter analysis uncovered multiple cis-acting elements, particularly those responsive to light and hormones. In addition, we examined the expression profiling of IbC5-MTases and IbdMTases genes during storage root development, revealing that several were highly expressed during the early and rapid expansion stages. These findings suggest that C5-MTases and dMTases may contribute to the regulation of storage root formation in sweet potato through epigenetic mechanisms, offering valuable insights for future functional studies and epigenetic breeding efforts.

## 1. Introduction

DNA methylation is a widespread epigenetic modification found in a diverse range of organisms, including bacteria, plants, and animals [1,2,3]. It refers to chemical changes to DNA or its associated proteins that influence gene expression without altering the underlying DNA sequence. In plants, DNA methylation predominantly occurs in symmetrical CG and CHG sequence contexts, as well as in the asymmetrical CHH context (where H = A, T, or C), which can be maintained or established de novo through distinct pathways [4]. Methylation in the CG and CHG contexts is generally maintained by methyltransferase 1 (MET1) and chromomethylase 2/3 (CMT2/3). In contrast, methylation in the CHH context is primarily maintained by CMT2 and domain rearranged methyltransferase 2 (DRM2) via the DNA methylation pathway directed by 24-nt small interfering RNAs [5,6]. DNA methylation dynamics are mediated by the coordinated activities of DNA methyltransferases, which establish and maintain methylation, and DNA demethylases, which actively remove methyl groups from DNA to maintain epigenetic plasticity [4,7].

Beyond the regulation of gene expression, DNA methylation plays critical roles in transposon repression, gene transcriptional regulation, genomic imprinting, and gene silencing [4]. Establishment, maintenance, and removal of DNA methylation are closely linked to fundamental processes such as plant growth and development [8], as well as secondary metabolism [9]. DNA methylation dynamics are orchestrated by two opposing groups of enzymes: DNA methyltransferases (C5-MTases), which catalyze the addition of methyl groups, and DNA demethylases (dMTases), which remove methylation marks to maintain epigenetic plasticity.

C5-MTases in plants can be classified into four subgroups based on the structural domains: DNA methyltransferase 1 (MET1), chromomethylase (CMT), domains rearranged methyltransferase (DRM), and DNA methyltransferase 2 (DNMT2) [10,11,12]. MET1 is responsible for the maintenance of CG methylation during DNA replication, while CMT3 and CMT2 are involved in maintaining CHG and CHH methylation, respectively [13]. DRM2, a key enzyme in the RNA-directed DNA methylation (RdDM) pathway, catalyzes de novo methylation in both symmetric and asymmetric contexts [14]. Conversely, DNA demethylation in plants can occur through passive or active mechanisms. Passive demethylation results from the failure to maintain DNA methylation during replication, whereas active demethylation is catalyzed by DNA glycosylase enzymes, which initiate base excision repair pathways [15,16]. In *Arabidopsis thaliana*, four major DNA demethylases have been identified: DEMETER (DME), REPRESSOR OF SILENCING 1 (ROS1), DEMETER-LIKE 2 (DML2), and DEMETER-LIKE 3 (DML3) [17]. These enzymes directly remove methylated cytosines, thereby providing a mechanism to reverse epigenetic silencing and regulate gene expression dynamically during development and environmental responses.

Extensive research has characterized C5-MTase and dMTase genes in a variety of plant species, including *Arabidopsis thaliana* [18], *Oryza sativa* [19], tomato [20], and *Cyclocarya paliurus* [21]. In the tea plant (*Camellia sinensis*), DNA methyltransferase and demethylase gene families were found to contain multiple cis-acting elements associated with hormone responses, plant growth, development, and abiotic stress signaling [22]. In *Ricinus communis*, genome-wide analysis identified eight DNA methyltransferase genes and three DNA demethylase genes, showing high conservation with other plant species and harboring cis-elements involved in osmotic stress responses [23].

While DNA methylation has been extensively studied in the context of plant responses to abiotic stress [24,25], accumulating evidence indicates that it also plays a vital role in developmental regulation, particularly in organ formation and differentiation [16,26]. For instance, in *Arabidopsis thaliana*, changes in DNA methylation patterns are essential for the transition from vegetative to reproductive growth [27,28]. In tomato, the regulation of fruit ripening involves dynamic DNA demethylation of ripening-related gene promoters, mediated by DML-type demethylases [26,29]. The mechanisms underlying DNA hypomethylation during the ripening of tomato and strawberry are distinct. In contrast to tomatoes, DNA demethylase genes are not upregulated during strawberry ripening. Instead, the ripening process in strawberries is associated with a downregulation of genes involved in RNA-directed DNA methylation (RdDM). This suppression of the RdDM pathway leads to genome-wide DNA hypomethylation, thereby facilitating the progression of fruit ripening in strawberries [30]. Furthermore, a study on root crops such as potato tuber [31] has suggested that epigenetic modifications contribute to storage root development.

Sweet potato (*Ipomoea batatas* L.) is a globally important tuberous crop, known for its rich nutritional profile, stress tolerance, and adaptability to marginal soils. In Africa, it serves as a staple food and a key source of calories, contributing to food security and rural livelihoods [32]. The storage root of sweet potato, formed through the secondary thickening of adventitious roots, is the main edible and economic organ. Storage root formation is a complex developmental process involving cell division, cell expansion, vascular cambium activation, and starch accumulation [33]. Despite its significance, the epigenetic mechanisms underlying sweet potato development and adaptation remain largely unexplored. Given the evolutionary relationship between diploid *Ipomoea trifida* (a putative ancestor) and cultivated hexaploid *I. batatas*, comparative analysis of methylation-related gene families in both genomes provides valuable insight into the retention and divergence of epigenetic regulators following polyploidization. In the present study, we performed a genome-wide identification and characterization of the C5-MTase and dMTase gene families in diploid wild sweet potato (*Ipomoea trifida* ‘Y22’) and autohexaploid cultivated sweet potato (*Ipomoea batatas* ‘Nancy Hall’). We analyzed their gene structures, conserved motifs, chromosomal distributions, syntenic relationships, and cis-acting regulatory elements. Furthermore, we examined the expression profiles of these genes during four critical stages of sweet potato storage root development, aiming to elucidate their potential roles in regulating storage root initiation and expansion. Our findings provide the first comprehensive overview of DNA methylation-related enzymes in sweet potato and suggest that dynamic epigenetic regulation mediated by C5-MTases and dMTases may play crucial roles during storage root development. This work lays the foundation for future functional studies and offers novel perspectives for sweet potato genetic improvement through epigenetic manipulation.

## 2. Results

### 2.1. Genome-Wide Identification of C5-MTase and dMTase Genes in Sweet Potato

To identify the C5-MTase and dMTase gene family members in sweet potato, we employed a hybrid approach combining BLASTP and HMM algorithms to analyze the sweet potato genome. This analysis led to the identification of 16 C5-MTase and 10 dMTase genes across the two sweet potato genomes. In total, eight C5-MTase and five dMTase genes were identified in the diploid genome, and a comparable number (eight IbC5-MTase and five IbdMTase) was found in the autohexaploid genome. In the diploid genome, the ItC5-MTase genes included three MET (*ItMET1/2/3*), two CMT (*ItCMT2/3*), and three DRM (*ItDRM1/2/3*). In the autohexaploid genome, the IbC5-MTase genes consisted of three MET (*IbMET1/2/3*), two CMT (*IbCMT2/3*), and three DRM (*IbDRM1/2/3*). The protein lengths of the 16 C5-MTase genes ranged from 295 amino acids (*ItDMR1*) to 2161 amino acids (*IbMET3*), with predicted molecular weights spanning from 33,366.13 to 243,939.95 Da. The isoelectric point (pI) values ranged from 4.85 to 8.91 (Table 1 and Appendix A). The calculated grand average of hydropathicity (GRAVY) values ranged from −0.571 to −0.257, indicating that all C5-MTase proteins are hydrophilic.

A total of 10 dMTase proteins were identified, consisting of 2 ROS (*ItROS1a/b*), 1 DML (*ItDML3*), and 2 DME (*ItDME1/2*) in the diploid species, and 2 ROS (*IbROS1a/b*), 1 DML (*IbDML3*), and 2 DME (*IbDME1/2*) in the autohexaploid species. Sequence characteristic analysis revealed that the protein lengths of dMTase varied from 1721 amino acids (*IbDML3*) to 2004 amino acids (*ItDME2*). The predicted molecular weights ranged from 191,203.84 Da to 223,396.42 Da, while the isoelectric points (pI) spanned from 5.84 to 8.31. All proteins had GRAVY (grand average of hydropathy) values below 0, generally hydrophilic with varying isoelectric points, some of which are above seven and hydrophilic in nature (Table 1).

### 2.2. Phylogenetic Analysis of C5-MTase and dMTase in Sweet Potato and Other Plant Species

To further investigate the evolutionary relationship between C5-MTase and dMTase, a phylogenetic tree was constructed using the protein sequences of C5-MTase and dMTase from *Arabidopsis thaliana* (Figure 1). The phylogenetic trees clearly separated the sweet potato MTases into major clusters corresponding to the known CMT, MET, and DRM/DNMT2 subfamilies (Figure 1A). Similarly, the dMTases clustered into distinct ROS1, DME, and DML subfamilies (Figure 1B). Additionally, the protein sequences of C5-MTase and dMTase showed high conservation across diploid and autohexaploid sweet potato as well as *A. thaliana*, suggesting their functional similarity across these species.

### 2.3. Chromosomal Location of IbC5-MTase and IbdMTase Genes

We determined the physical locations of the identified *IbC5-MTase* and *IbdMTase* genes on the pseudochromosomes of the diploid wild sweet potato Y22 and the haplotype pseudochromosomes of hexaploid cultivated sweet potato ‘Nancy Hall’ (Figure 2). In the diploid genome, the 8 *ItC5-MTase* genes were distributed across multiple pseudochromosomes, including Itr2xChr02, Itr2xChr04, Itr2xChr07, Itr2xChr10, Itr2xChr11, Itr2xChr12, Itr2xChr13, and Itr2xChr14 (Figure 2A). The five *ItdMTase* genes were located on Itr2xChr02, Itr2xChr04, Itr2xChr07, Itr2xChr08, and Itr2xChr13 (Figure 2A). The distribution was visibly uneven, with some pseudochromosomes containing multiple genes while others had none among those identified.

In the hexaploid cultivated sweet potato genome (Figure 2B), the eight *IbC5-MTase* genes were dispersed on IbaNHchr02, IbaNHchr04, IbaNHchr07, IbaNHchr10, IbaNHchr11, IbaNHchr12, and IbaNHchr14. The five *IbdMTase* genes were mapped to IbaNHchr02, IbaNHchr04, IbaNHchr07, IbaNHchr08, IbaNHchr13, and IbaNHchr14 (Figure 2B). Similarly to the diploid genome, the genes in the hexaploid sweet potato were also unevenly distributed among the analyzed pseudochromosomes. A comparison of the gene locations between the two ploidy levels revealed instances where orthologous genes maintained their relative positions on corresponding chromosomes or pseudochromosomes, such as the clustering of MET genes on Itr2xChr04 and IbaNHchr04 (Figure 2).

### 2.4. Synteny Analysis Between Diploid and Hexaploid Sweet Potato Genomes

To investigate the genomic distribution and conservation of the identified sweet potato MTase and dMTase gene families in the context of polyploidization, we performed a detailed synteny analysis between the genome of wild diploid sweet potato Y22 and the haplotype genome of hexaploid cultivated sweet potato ‘Nancy Hall’ (Figure 3). The analysis revealed extensive collinear regions shared between the two genomes, which underscore their close evolutionary relationship and the preservation of large-scale genomic structures.

Importantly, we found that the majority of the identified *IbC5-MTase* and *IbdMTase* genes were located within these conserved collinear blocks. Syntenic pairs of these specific genes between the diploid and hexaploid genomes are clearly highlighted by red lines in Figure 3. The presence of these genes in these syntenic regions strongly suggests that many duplicates arising from ancestral polyploidization events were successfully retained in their original genomic contexts, contributing to the gene content of the hexaploid sweet potato.

However, the synteny map also graphically illustrated the complex genomic architecture resulting from polyploidization in the hexaploid sweet potato. We observed intricate collinear relationships, including prominent “many-to-many” connections (e.g., a region on a diploid chromosome segment corresponding to multiple regions on different hexaploid pseudochromosomes), a pattern indicative of whole-genome duplication (WGD) events followed by extensive genomic rearrangements, such as segmental duplications, translocations, and gene fractionation.

### 2.5. Conserved Motif and Gene Structure Analysis

To investigate the diversification of C5-MTase and dMTase proteins, the MEME tool was used to identify conserved motifs within these proteins. In C5-MTase proteins, up to eight motifs were identified with varying combinations present among different family members (Figure 4A), with MET1, CMT2, and DRM3 containing all motifs. Notably, motifs 1, 2, and 4 were the major conserved motifs in the DRM subfamilies. On the contrary, all dMTase proteins contain the same motif structure (Figure 4B). This conserved motif composition underscores a shared motif profile within dMTase proteins, highlighting their unique structural and functional characteristics. It can be inferred that the motifs shared between C5-MTase and dMTase proteins are likely associated with conserved biological functions, whereas motifs specific to certain proteins may contribute to gene-specific roles.

To further understand the evolution of these gene families, we analyzed the exon-intron structures of these genes. The analysis revealed varying exon numbers across their members. As shown in Figure 4, different exon-intron distribution patterns were observed within the C5-MTase gene family. The intron numbers ranged from 5 to 21 in ItC5-MTase, 14–18 in ItdMTase, 5–20 in IbC5-MTase, and 14–18 in IbdMTase. Among them, *ItCMT3* had the largest number of introns (21), while *IbMET2* contained only five introns.

### 2.6. Cis-Acting Element Analysis of C5-MTase and dMTase

To predict the potential transcriptional regulatory mechanisms governing the expression of the identified sweet potato C5-MTase and dMTase genes, we analyzed the presence of putative cis-acting regulatory elements within their upstream 2000 bp promoter sequences (Figure 5). This analysis revealed that the promoter regions of both It/IbC5-MTase (Figure 5A) and It/IbdMTase (Figure 5B) genes were highly enriched with various types of cis-acting elements, indicating complex transcriptional regulation by a multitude of factors. A detailed list of the identified elements, including their specific types, positions, and corresponding regulatory functions, is provided in Appendix A.

We found widespread presence of elements related to light responsiveness (e.g., G-box, GT1-motif, Box 4), suggesting that the expression of these genes may be influenced by light cues. Furthermore, a diverse array of plant hormone-responsive elements was predicted, including those associated with gibberellin (e.g., GARE-motif, P-box), jasmonic acid (e.g., MeJA-responsiveness, TGACG-motif, CGTCA-motif), abscisic acid (e.g., ABRE), auxin (e.g., AuxRR-core, TGA-element), and salicylic acid (e.g., TCA-element) (Figure 5). This strong presence of hormone-responsive elements highlights the potential for these genes to be regulated by plant hormones, which are central regulators of plant growth, development, and stress responses. Additionally, numerous elements linked to stress responses (e.g., TC-rich repeats, MBS, LTR, ARE) and developmental processes (e.g., CAT-box, meristem expression, seed-specific regulation) were also frequently identified (Figure 5). The presence of this rich collection of diverse cis-acting elements collectively suggests that the expression of sweet potato MTase and dMTase genes is finely tuned by a combination of environmental signals, hormonal cues, and developmental programs.

### 2.7. Expression Patterns of IbC5-MTase and IbdMTase Genes During Storage Root Development

To investigate the functional roles of *IbC5-MTase* and *IbdMTase* genes in sweet potato storage root development, transcript abundance was analyzed across four key developmental stages in the cultivar ‘Chuanshu228’: Stage 0 (25 DPT, non-swollen roots), Stage 1 (55 DPT, early thickening), Stage 2 (85 DPT, rapid bulking), and Stage 3 (120 DPT, maturation). Initial RNA-seq analysis (Figure 6A, Appendix A) revealed diverse expression profiles among these genes. With certain genes, such as *IbMET2* and *IbDME1,* consistently exhibiting low transcript levels, others displayed dynamic, stage-specific patterns. For example, *IbDME2* peaked at Stage 1, whereas *IbMET3* and *IbDRM2* reached their highest expression at Stage 2. In contrast, *IbROS1b*, *IbMET1*, and *IbROS1a* showed elevated transcript levels at Stage 0, suggesting potential involvement in early root formation.

To validate and further quantify these expression patterns, six representative genes (*IbMET1*, *IbCMT2*, *IbCMT3*, *IbDRM1*, *IbROS1a*, and *IbDML3*) were selected for quantitative real-time PCR (qRT-PCR) analysis (Figure 6B). The qRT-PCR results confirmed distinct temporal expression trends. *IbMET1*, *IbCMT*2, and *IbROS1a* exhibited the highest transcript levels at Stage 0, with expression declining across later stages, although *IbMET1* showed a secondary peak at Stage 2. *IbCMT3* also peaked at Stage 0, declined at Stages 1 and 3, but increased at Stage 2. In contrast, *IbDML3* maintained relatively high and stable expression from Stage 0 through Stage 2, peaking at Stage 2, before declining sharply at Stage 3. The combined analyses underscore the complex and stage-specific transcriptional regulation of *IbC5-MTase* and *IbdMTase* genes during sweet potato storage root development.

## 3. Discussion

DNA methylation, a pivotal epigenetic modification, plays a crucial role in regulating gene expression, genomic stability, and developmental processes in plants [4,10]. The dynamic balance of DNA methylation is maintained by cytosine-5 DNA methyltransferases (C5-MTases) and DNA demethylases (dMTases). In this study, we performed a comprehensive genome-wide analysis of the C5-MTase and dMTase gene families in both diploid wild sweet potato (*Ipomoea trifida*) and autohexaploid cultivated sweet potato (*Ipomoea batatas* ‘Nancy Hall’). Our findings provide new insights into the molecular characteristics and potential functional roles of these genes in sweet potato, particularly during storage root development. We identified 16 C5-MTase and 10 dMTase genes across the two sweet potato genomes, showing comparable numbers between the diploid and hexaploid species. The classification of these genes into MET, CMT, and DRM subfamilies for C5-MTases, and ROS1, DME, and DML3 subfamilies for dMTases, is consistent with previous reports in other plant species such as *Arabidopsis thaliana* [18], tomato [20], and rice [19]. Phylogenetic analysis further demonstrated that sweet potato C5-MTases and dMTases are highly conserved and cluster closely with their *Arabidopsis* counterparts, suggesting evolutionary conservation of function. The identification of these genes provides a foundation for understanding their roles in sweet potato growth, development, and stress responses.

Polyploidization is a major driver of genome evolution, often leading to gene family expansion and functional diversification [34,35]. However, our comparative analysis of the diploid wild sweet potato (*Ipomoea trifida*) and the cultivated autohexaploid sweet potato (*Ipomoea batatas*) revealed a striking conservation in both gene number and chromosomal positioning of DNA methylation-related genes. Both genomes harbor eight C5-MTase and five dMTase genes, with clear positional conservation—for instance, *ItMET2* and *ItMET3* are located on Itr2xChr04 in the diploid genome, while their hexaploid counterparts, *IbMET2* and *IbMET3*, are mapped to IbaNHchr04, suggesting that polyploidization did not disrupt their genomic locations (Figure 2). Synteny analysis further confirmed that most C5-MTase and dMTase genes were located within conserved collinear blocks between the two genomes, underscoring their evolutionary stability (Figure 3). The stable gene number and conserved genomic context indicate that the autohexaploid nature of sweet potato did not lead to dramatic expansion of these gene families, consistent with previous findings that epigenetic regulatory genes tend to be selectively retained or lost following polyploidization to maintain genome stability [30,36]. Moreover, although gene number is conserved, observed differences in expression profiles suggest subfunctionalization or neofunctionalization of methylation-related genes in the hexaploid genome. This pattern, also reported in other polyploid species, such as *Cyclocarya paliurus*, where whole-genome duplication contributed to gene diversification and structural genomic complexity [37], highlights both the evolutionary constraint and functional innovation of epigenetic regulators following polyploidization.

Analysis of gene structures and conserved motifs revealed that sweet potato C5-MTase and dMTase proteins exhibit similar motif architectures within their respective subfamilies, reflecting a high degree of functional conservation. This pattern is consistent with the observations in *C. paliurus* [21]. For instance, DMR3, CMT3, CMT2, and MET1 proteins shared several common motifs, whereas all dMTase proteins contained an identical set of conserved motifs. Gene structure analysis further showed variability in intron numbers among different subfamilies. For example, *ItCMT3* possessed 21 introns—the highest among the C5-MTases—whereas *IbMET2* contained only five introns. Interestingly, while C5-MTase genes displayed considerable variability in their exon–intron organization, the dMTase genes exhibited relatively conserved gene architectures, suggesting that active DNA demethylation processes may be under stronger evolutionary constraints to preserve enzymatic functionality [12]. This structural diversity among C5-MTases likely reflects subfamily-specific evolutionary trajectories, with MET genes showing greater intron number stability compared to DRMs. Moreover, the conservation of exon–intron structures within subfamilies across diploid and hexaploid sweet potato species further supports the notion of functional conservation throughout evolution.

Additionally, cis-acting elements in the promoter regions are essential for the control of gene expression, often mediating responses to various developmental and environmental signals [38]. Notably, light-responsive elements were predominantly enriched across nearly all analyzed genes, suggesting that light signaling plays a major role in regulating the transcription of DNA methylation-related genes. This observation is consistent with findings in NF-Y gene families in sweet potato [39]. Given the critical role of light in root development and carbohydrate metabolism, such a regulatory mechanism may have profound implications for storage root formation [31]. In addition to light-responsive elements, numerous hormone-responsive motifs, including those associated with abscisic acid (ABA), gibberellins (GA), jasmonic acid (JA), salicylic acid (SA), and auxin. These findings suggest that sweet potato C5-MTase and dMTase genes may function at the intersection of hormonal signaling and epigenetic regulation, potentially influencing key biological processes such as growth, development, and stress adaptation [40]. Additionally, recent studies have highlighted the role of strigolactones—an important class of plant hormones involved in root development and stress responses—in epigenetic regulation. Strigolactone signaling may intersect with DNA methylation pathways to modulate root system architecture, particularly under nutrient-limited conditions [41]. The predicted presence of strigolactone-responsive elements in some IbC5-MTase promoter regions supports the hypothesis that DNA methylation might integrate hormonal and developmental cues. Furthermore, the prevalence of motifs related to meristem activity and seed-specific regulation underscores the involvement of these genes in organogenesis. Despite overall similarities in promoter architecture, subtle differences were observed between C5-MTase and dMTase genes. For instance, certain dMTase promoters exhibited a higher density of ABA-responsive and stress-related elements, hinting at distinct regulatory mechanisms for DNA demethylation compared to DNA methylation during storage root development.

RNA-seq and qRT-PCR analyses revealed dynamic expression patterns of IbC5-MTase and IbdMTase genes across four stages of sweet potato storage root development, suggesting stage-specific functions (Figure 6). Notably, the peak expression of *IbMET1*, *IbCMT2*, and *IbROS1a* at Stage 0 implies their involvement in early epigenetic reprogramming. Such reprogramming mechanisms are crucial during the transition from adventitious roots to storage roots, as has been documented for various developmental shifts in sweet potato [42]. This early activity may further relate to the regulation of genes controlling cell fate and cambium activity. Indeed, the precise control of cell identity and meristematic function is fundamental to organogenesis, and epigenetic factors are known to play a role in these processes. The presence of multiple hormone-responsive cis-elements (e.g., auxin-responsive) in their promoters (Figure 5) supports their potential role as targets of hormonal cues initiating storage root formation. This is consistent with the established understanding that hormonal signals, particularly auxin, are key initiators of root development and often intersect with epigenetic regulatory pathways [43,44]. In contrast to these early-peaking genes, *IbDML3* exhibited sustained high expression from Stage 0 to Stage 2, peaking at Stage 2. This expression pattern suggests a key role for IbDML3-mediated demethylation during phases of active cell proliferation and early root bulking. Active DNA demethylation by DML family proteins is frequently associated with the activation of genes required for growth and development in various plant systems [45]. Interestingly, *IbDRM1* displayed minimal expression at Stage 0, but its transcripts accumulated significantly at Stage 3. This late-stage induction implicates *IbDRM1* in maturation-related processes, possibly through RNA-directed DNA methylation (RdDM), which could fine-tune gene expression associated with starch biosynthesis or contribute to long-term epigenetic stabilization as the storage organ matures. The RdDM pathway is well-characterized for its function in establishing de novo methylation and silencing, often playing roles in later developmental stages or stress responses [46,47]. Meanwhile, consistently low expression of *IbMET2* and *IbDME1* may reflect basal or redundant functions. These expression trends, together with promoter analysis, offer a framework for future functional validation, particularly for genes with strong stage-specific activity during thickening. To improve the robustness of transcriptomic findings, future studies should validate functional characterization, especially those showing strong expression shifts during root thickening stages. Overall, the findings emphasize the tight linkage between epigenetic regulation and storage root development in sweet potato, aligning with broader observations in plant root epigenetics [48].

## 4. Materials and Methods

### 4.1. Identification of the C5-MTase and dMTase Genes in Sweet Potato

To identify C5-MTase and dMTase genes in the sweet potato genome, a multi-step approach was employed. First, genome sequences of both diploid (*Ipomoea trifida*) and autohexaploid (*Ipomoea batatas*) sweet potato were obtained. Known C5-MTase and dMTase protein sequences from *Arabidopsis thaliana* were used as queries for a BLASTP search against the sweet potato genome database, with an e-value cutoff of <1 × 10^−5^ to identify homologous genes. Additionally, Hidden Markov Model (HMM) profiles for the C5-MTase (PF00145) and dMTase (PF00730, PF15628) domains were retrieved from the Pfam database [49]. Candidate proteins were further identified using the HMMER hmmsearch tools with an e-value threshold of <1 × 10^−3^. The presence of conserved domains in candidate proteins was verified using the Conserved Domains Database (CDD, https://www.ncbi.nlm.nih.gov/Structure/, accessed on 10 January 2025). Protein sequence length, predicted molecular weight, and theoretical isoelectric point (pI) were determined via the ExPASy server [50].

### 4.2. Sequence Alignment and Phylogenetic Analysis

To explore the evolutionary relationships between sweet potato and *Arabidopsis* C5-MTase and dMTase proteins, full-length sequences were aligned using MUSCLE. Phylogenetic trees were constructed in MEGA11 [51] using the Maximum Likelihood (ML) method with the Jones–Taylor–Thornton (JTT) model, Gamma Distributed (G) rate variation, partial deletion of gaps, and 1000 bootstrap replicates. Tree visualization was performed with Evolview v3 [52].

### 4.3. Gene Structures, Conserved Motif, and Domain Analysis

The exon-intron organization of the identified C5-MTase and dMTase genes was analyzed and visualized using TBtools (v2.210) software [53]. Conserved motifs within the protein sequences were predicted using the MEME Suite online server [54], with the number of motifs set to eight and default settings. Conserved domains were identified using the Batch CD-Search tool (NCBI), and all results were integrated and visualized using TBtools.

### 4.4. Chromosomal Distribution and Gene Duplication

Chromosomal locations of the C5-MTase and dMTase genes were retrieved from the GFF3 annotation files of the sweet potato genomes and visualized using TBtools (v2.210) [53]. Gene duplication events, including segmental and tandem duplications, were identified using the Multiple Collinearity Scan toolkit (MCScanX) [55]. Synteny relationships between *I. batatas* and *I. trifida* C5-MTase/dMTase genes were further examined and visualized using the “Multiple Synteny Plot” function in TBtools.

### 4.5. Cis-Acting Element Analysis for C5-MTase and dMTase Promoters

To investigate potential transcriptional regulation, 2000 bp upstream promoter sequences of each *I. batatas* C5-MTase and dMTase genes were extracted. The PlantCARE database [56] was used to identify and classify the cis-acting regulatory elements. Motifs associated with plant growth, development, phytohormone responsiveness, and stress responses were analyzed.

### 4.6. RNA Extraction and Sequencing

The sweet potato (*Ipomoea batatas*) cultivar ‘Chuanshu 228’ was used for transcriptome analysis. Plants were grown in pots under greenhouse conditions at the Crop Research Institute, Sichuan Academy of Agricultural Sciences, Chengdu, China. Storage root samples for RNA-seq were collected at four key developmental stages: Stage 0 (25 days post-transplanting, non-swollen adventitious roots), Stage 1 (55 days, early thickening), Stage 2 (85 days, rapid bulking), and Stage 3 (120 days, maturation with starch accumulation). For each time point, three independent biological replicates were harvested. All samples were immediately frozen in liquid nitrogen and stored at −80 °C until RNA extraction. Total RNA was extracted from sweet potato storage root using the Plant RNA Easy Fast RNA extraction kit (TIANGEN Biotech (Beijing) Co., Ltd., Beijing, China)), following the manufacturer’s instructions. RNA concentration and purity were assessed with a NanoDrop 2000 spectrophotometer (Thermo Fisher Scientific, Wilmington, DE, USA), and RNA integrity was evaluated using an Agilent 2100 Bioanalyzer using the RNA Nano 6000 Assay Kit (Agilent Technologies, Santa Clara, CA, USA). Only RNA samples with an RNA Integrity Number (RIN) > 9 were used for cDNA library construction. Sequencing was conducted on the Illumina HiSeq 4000 (Illumina, Inc., San Diego, CA, USA) platform to generate paired-end reads.

### 4.7. Transcriptome Analysis

Raw sequencing reads were initially subjected to quality control using fastp, which removed adapter sequences and filtered out reads containing poly-N regions (>10%) or those with low-quality bases. The resulting clean reads were then aligned to the sweet potato genome using HISAT2 v2.2.1 with default settings [57]. Transcript assembly and quantification were performed using StringTie v2.2.1 [58], and gene expression levels were calculated in Fragments Per Kilobase of exon model per Million mapped fragments (FPKM). Differentially expressed genes (DEGs) were identified using DESeq2 v1.44.0 [59], based on a negative binomial distribution model with a threshold of fold change (FC) ≥ 2 and a false discovery rate (FDR) < 0.01.

### 4.8. qRT-PCR Validation of IbC5-MTase and IbdMTase Gene Expression

To validate the RNA-seq results, quantitative real-time PCR (qRT-PCR) was conducted using a subset of the total RNA samples previously extracted from ‘Chuanshu228’ storage roots at the four developmental stages (as described in Section 4.6). First-strand cDNA was synthesized using a PrimeScript™ RT Reagent Kit (Takara Bio (Dalian) Co., Ltd., Dalian, China). qRT-PCR was performed on a CFX96 Real-Time PCR System (Bio-Rad Laboratories, Inc., Hercules, CA, USA) using SYBR^®^ Green Master Mix (Takara Bio (Dalian) Co., Ltd., Dalian, China), following established protocols and including a melting curve analysis [53]. Gene-specific primers are listed in Appendix A, with IbACTIN7 serving as the internal reference gene. Relative expression levels were calculated using the 2^−ΔΔCt^ method based on three biological replicates, and statistical significance was determined using Tukey’s HSD (*p* < 0.05).

## 5. Conclusions

In summary, we systematically identified and characterized the C5-MTase and dMTase gene families in both diploid and hexaploid sweet potato genomes. Phylogenetic analysis, structural characterization, and synteny mapping revealed high conservation of these genes, suggesting crucial roles in sweet potato genome regulation. Cis-acting element analysis indicated that their expression may be finely tuned by multiple developmental and environmental cues. Importantly, expression profiling during storage root development demonstrated that several *C5-MTase* and *dMTase* genes are dynamically regulated, implying their involvement in critical processes such as secondary growth, vascular development, and biomass accumulation. These findings highlight the potential importance of epigenetic regulation in storage root formation. Future research should focus on functional validation of candidate genes and elucidation of the underlying regulatory networks integrating DNA methylation, hormonal control, and metabolic pathways. Such efforts may facilitate the development of novel sweet potato varieties with improved storage root yield, nutritional quality, and stress resilience through epigenetic breeding strategies.

## Figures and Tables

**Figure 1 plants-14-01735-f001:**
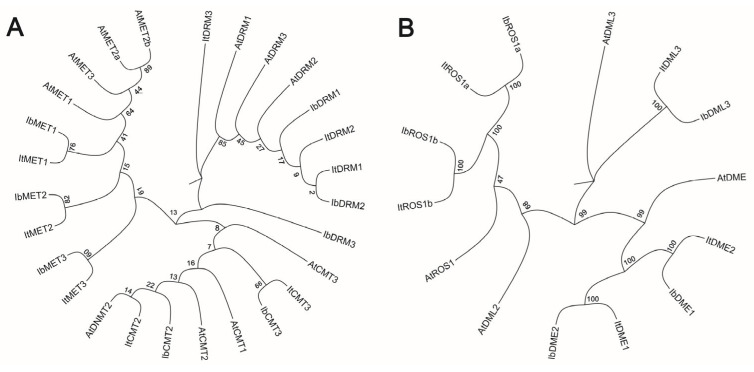
Phylogenetic analysis of sweet potato MTases and dMTases. (**A**) Phylogenetic tree of C5-MTase proteins from diploid *Ipomoea trifida* (*It*), hexaploid *Ipomoea batatas* (*Ib*), and *Arabidopsis thaliana* (*At*). (**B**) Phylogenetic tree of dMTase proteins from It, Ib, and At. Branch support was assessed using bootstrap values based on 1000 replicates.

**Figure 2 plants-14-01735-f002:**
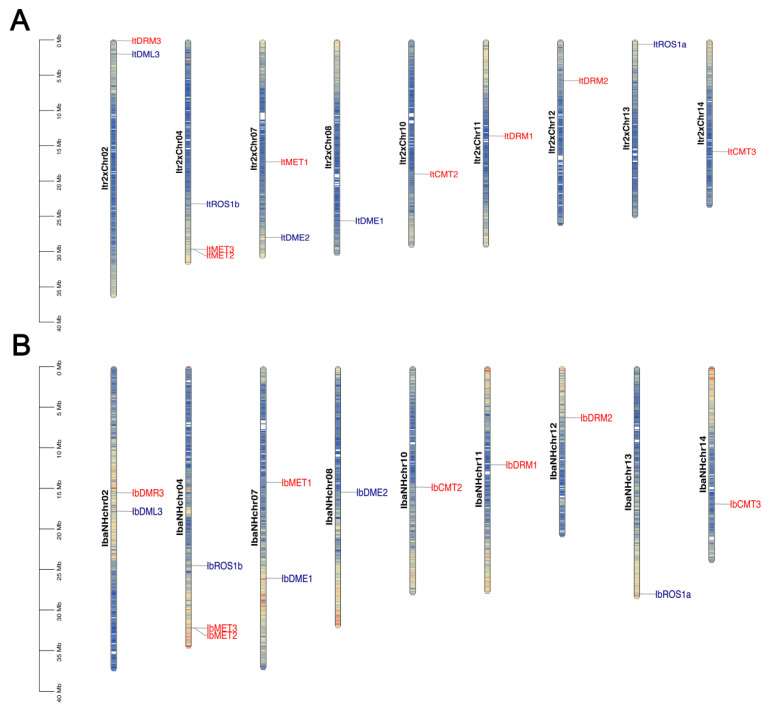
Chromosomal location of *IbC5-MTase* and *IbdMTase* genes. (**A**) Distribution of ItC5-MTase (red) and ItdMTase (blue) genes on diploid *Ipomoea trifida* Y22 pseudochromosomes. (**B**) Distribution of IbC5-MTase (red) and IbdMTase (blue) genes on hexaploid *Ipomoea batatas* haplotype pseudochromosomes. The color gradient along each chromosome bar represents gene density, where blue indicates high gene density and orange/yellow indicates low gene density.Scale bars indicate approximate physical distance (Mb).

**Figure 3 plants-14-01735-f003:**
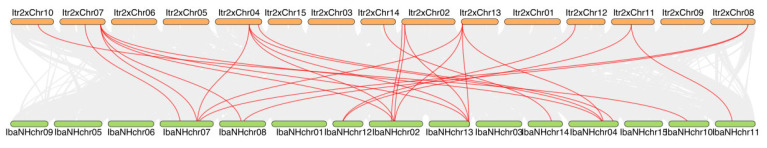
Synteny analysis between diploid and hexaploid sweet potato genomes. Collinearity analysis between diploid and hexaploid genomes. Gray lines show general collinear regions; red lines highlight syntenic C5-MTase and dMTase gene pairs.

**Figure 4 plants-14-01735-f004:**
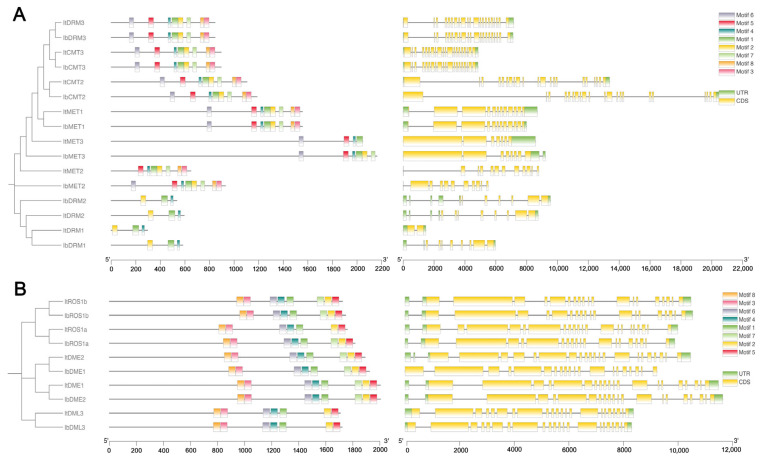
Gene structure and conserved motif analysis of It/IbC5-MTase (**A**) and It/IbdMTase (**B**) families. Left diagrams show phylogenetic trees and conserved protein motifs (scale bar: protein length, aa). Right diagrams depict gene exon-intron structures (scale bar: DNA length, bp).

**Figure 5 plants-14-01735-f005:**
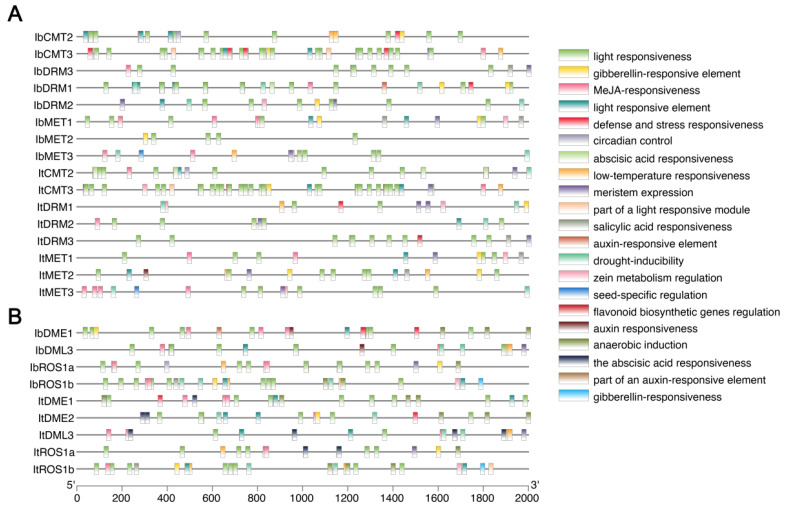
Cis-acting element analysis in promoter regions of sweet potato MTase and dMTase genes. (**A**) Predicted cis-acting elements in the 2000 bp upstream regions of IbC5-MTase and ItC5-MTase genes. (**B**) Predicted cis-acting elements in the 2000 bp upstream regions of IbdMTase and ItdMTase genes. The scale bar at the bottom indicates the length of the promoter sequence in base pairs (bp).

**Figure 6 plants-14-01735-f006:**
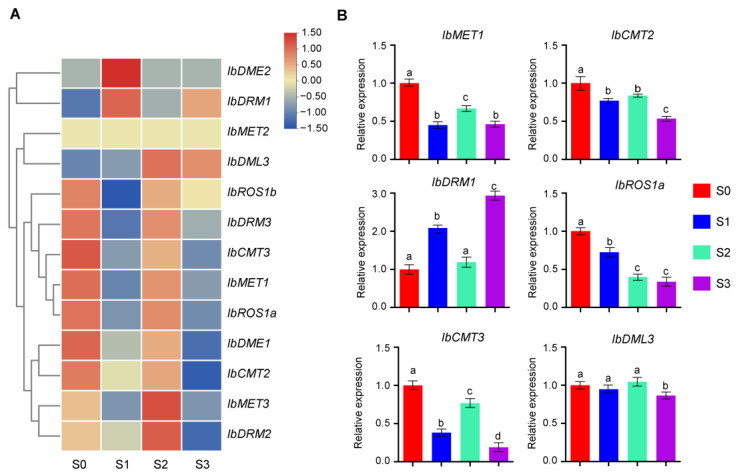
Expression patterns of *IbC5-MTase* and *IbdMTase* genes during storage root development. (**A**) Heatmap showing expression levels of selected genes across four storage root developmental stages (Stage 0–3). (**B**) Relative expression (qRT-PCR) of *IbMET1*, *IbCMT2*, *IbCMT3*, *IbDRM1*, *IbROS1a*, and *IbDML3* across four developmental stages (S0–S3). Data are mean ± SD (*n* = 3). Different letters above the bars indicate significant differences (Tukey’s HSD, *p* < 0.05).

**Table 1 plants-14-01735-t001:** Basic information of C5-MTase and dMTase genes in diploid and autohexaploid sweet potatoes.

Gene Name	Gene ID	ORF (bp)	Amino Acid (aa)	Molecular Weight (Da)	pI	Instability Index	Aliphatic Index	GRAVY Value
Cytosine-5 DNA methyltransferases in diploid								
*ItMET1*	Itr2xGene015493	4665	1555	175,114.63	5.75	41.63	74.93	−0.514
*ItMET2*	Itr2xGene009786	1941	647	72,644.74	8.91	42.44	77.3	−0.472
*ItMET3*	Itr2xGene009785	6138	2046	231,505.64	6.23	48.17	82.97	−0.37
*ItCMT2*	Itr2xGene021773	3312	1104	124,150.65	5.76	57.45	77.59	−0.552
*ItCMT3*	Itr2xGene030412	2682	894	99,407.08	5.1	36.39	79.04	−0.456
*ItDRM1*	Itr2xGene024374	885	295	33,366.13	7.75	44.48	95.76	−0.257
*ItDRM2*	Itr2xGene026296	1779	593	66,766.51	4.85	47.94	81.05	−0.443
*ItDRM3*	Itr2xGene002910	2532	844	95,713.01	5.12	43.39	76.92	−0.538
Demethylases in diploid								
*ItROS1a*	Itr2xGene027250	5277	1759	196,236.06	5.84	47.61	71.98	−0.67
*ItROS1b*	Itr2xGene008953	5169	1723	191,952.08	6.37	52.02	68.36	−0.761
*ItDML3*	Itr2xGene003152	5121	1707	191,203.84	8.17	53.73	72.37	−0.633
*ItDME1*	Itr2xGene018529	6009	2003	223,140.95	8.31	49.07	69.44	−0.724
*ItDME2*	Itr2xGene016302	5673	1891	210,000.1	7.56	46.96	69.79	−0.701
Cytosine-5 DNA methyltransferases in autohexaploid								
*IbMET1*	Iba6xGene015870	4665	1555	175,034.49	5.73	41.87	74.44	−0.521
*IbMET2*	Iba6xGene010270	2787	929	104,312.43	7.7	44.07	74.68	−0.448
*IbMET3*	Iba6xGene010269	6483	2161	243,939.95	6.13	48.29	84.07	−0.332
*IbCMT2*	Iba6xGene023300	3558	1186	133,423.13	6.14	58.99	76.75	−0.571
*IbCMT3*	Iba6xGene031894	2682	894	99,448.23	5.08	35.96	79.36	−0.448
*IbDRM1*	Iba6xGene025585	1746	582	65,522.3	4.92	45.39	79.4	−0.534
*IbDRM2*	Iba6xGene027558	1599	533	60,203.28	5.16	47.26	80.26	−0.454
*IbDRM3*	Iba6xGene004347	2532	844	95,683.91	5.12	42.77	77.04	−0.541
Demethylases in autohexaploid								
*IbROS1a*	Iba6xGene030318	5448	1816	202,863.41	5.69	48.71	71.28	−0.693
*IbROS1b*	Iba6xGene009389	5238	1746	194,639.05	6.56	51.81	68.02	−0.78
*IbDME1*	Iba6xGene016753	5769	1923	213,397.76	7.71	47.1	68.88	−0.713
*IbDME2*	Iba6xGene018826	6012	2004	223,396.42	8.31	50.12	69.41	−0.717
*IbDML3*	Iba6xGene004621	5163	1721	192,880.62	7.17	53.2	72.57	−0.63

Note: ORF: Open reading frame, pI: theoretical isoelectric point, GRAVY: grand average of hydrophobicity.

## Data Availability

The raw sequence data reported in this paper have been deposited in the Genome Sequence Archive (Genomics, Proteomics & Bioinformatics 2021) in National Genomics Data Center (Nucleic Acids Res 2022), China National Center for Bioinformation/Beijing Institute of Genomics, Chinese Academy of Sciences (GSA: CRA025737) that are publicly accessible at https://ngdc.cncb.ac.cn/gsa (accessed on 16 May 2025).

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
