# Peer review of "Genome-Wide Identification and Analysis of DNA Methyltransferase and Demethylase Gene Families in Sweet Potato and Its Diploid Relative"

_plants, 2025, doi:10.3390/plants14111735_

Round 1

Reviewer 1 Report

Comments and Suggestions for Authors

This manuscript titled "Genome-wide identification and analysis of DNA methyltransferase and demethylase gene families in sweet potato and its diploid relative" presents a genome-wide analysis of DNA methylation-related gene families in diploid and hexaploid sweet potato species. While the study follows a standard pipeline for gene family analysis—including gene identification, phylogenetics, motif structure, promoter analysis, and expression profiling—the overall novelty is limited. The manuscript is well-structured and written in generally clear English, although several grammatical and typographical issues should be addressed. The data presented are consistent and complete, but the biological interpretation lacks depth and could benefit from more contextual discussion regarding gene function and potential applications..

Major Comments:

  1. Although expression profiling was conducted during storage root development, the link between specific C5-MTase/dMTase genes and physiological processes is not clearly discussed. Including potential functional implications (e.g., specific developmental roles or regulatory mechanisms) would improve the impact of the work.
  2. The manuscript includes both diploid (I. trifida) and autohexaploid (I. batatas) sweet potato, but the reason for selecting both is not clearly explained.
  3. It is not entirely clear whether the RNA-seq data used in this study were generated by the authors or obtained from a public database. If newly generated, please clearly state this in the Methods section, and ensure that the raw sequencing data are deposited in a public repository (e.g., NCBI SRA) with accession numbers provided.
  4. The study examines gene expression at four developmental stages (Stage 0, 1, 2, and 3; corresponding to 25, 55, 85, and 120 days after transplanting). However, the rationale for choosing these specific time points is not explained. A brief justification based on developmental milestones or previous literature would help readers better understand the biological relevance of these stages.
  5. While the RNA-seq data provides useful insights into gene expression patterns during storage root development, it would significantly strengthen the study if a subset of key genes were validated using quantitative PCR (qPCR). This would help confirm the reliability of the transcriptomic data and support the conclusions drawn from expression analysis.

Minor Comments:

  1. In the reference list, Latin names of plant species should be italicized. Please carefully review the entire manuscript to ensure all Latin names are correctly formatted in italics. In addition, there is inconsistency in the capitalization of words in reference titles—for instance, words in the titles from Lines 521–522 are capitalized, while those in Lines 523–524 are not. Several references in the bibliography are missing journal names or include incomplete citation details (e.g., references [24], etc.). These inconsistencies indicate that the reference formatting requires thorough revision. Please carefully check all references and revise them according to the journal’s formatting requirements.
  2. The phylogenetic analysis is limited to Arabidopsis and lacks comparison with other crop species (e.g., potato, cassava).
Comments on the Quality of English Language

The overall quality of the English language in this article is generally well-written, yet there are certain aspects that necessitate refinement.

  1. Line 45, “through distinct pathway” → “through distinct pathways”.
  2. Line 215, “strucuture” → “structure”
  3. Line 215, “all dMTase proteins contains” → “all dMTase proteins contain”
  4. Line 358, “These findings suggests” → “These findings suggest”

Please meticulously review the entire text for the accuracy of English tenses and the fluidity of the language and proceed with the necessary revisions. 

Author Response

Reviewer1

This manuscript titled "Genome-wide identification and analysis of DNA methyltransferase and demethylase gene families in sweet potato and its diploid relative" presents a genome-wide analysis of DNA methylation-related gene families in diploid and hexaploid sweet potato species. While the study follows a standard pipeline for gene family analysis—including gene identification, phylogenetics, motif structure, promoter analysis, and expression profiling—the overall novelty is limited. The manuscript is well-structured and written in generally clear English, although several grammatical and typographical issues should be addressed. The data presented are consistent and complete, but the biological interpretation lacks depth and could benefit from more contextual discussion regarding gene function and potential applications.

Major Comments:

1.Although expression profiling was conducted during storage root development, the link between specific C5-MTase/dMTase genes and physiological processes is not clearly discussed. Including potential functional implications (e.g., specific developmental roles or regulatory mechanisms) would improve the impact of the work.

Response: Thank you for this constructive feedback. We acknowledge the importance of more explicitly linking the expression of specific IbC5-MTase/IbdMTase genes to potential physiological processes and regulatory mechanisms during storage root development. We have revised the Discussion section to provide a more detailed interpretation of our expression data (RNA-seq and qPCR)(P. 11-12, L. 398-423). We believe these additions provide a more direct and nuanced discussion of the potential functional implications of the observed expression patterns for specific gene members, thereby strengthening the biological interpretation of our findings.

2.The manuscript includes both diploid (I. trifida) and autohexaploid (I. batatas) sweet potato, but the reason for selecting both is not clearly explained.

Response: Thank you for your valuable comment. We have added an explanation at the last paragraph of the Introduction to clarify the rationale for analyzing both diploid and hexaploid sweet potato species(P. 3, L. 103-113). This comparative approach enables us to explore gene family conservation and divergence following polyploidization.

3.It is not entirely clear whether the RNA-seq data used in this study were generated by the authors or obtained from a public database. If newly generated, please clearly state this in the Methods section, and ensure that the raw sequencing data are deposited in a public repository (e.g., NCBI SRA) with accession numbers provided.

Response: Thank you for your suggestion. We have clarified in the Materials and Methods section that the RNA-seq data were generated. The raw sequencing data have been submitted to GSA under accession number CRA025737, which provided in Data Availability Statement section(P. 14, L. 532-536).

4.The study examines gene expression at four developmental stages (Stage 0, 1, 2, and 3; corresponding to 25, 55, 85, and 120 days after transplanting). However, the rationale for choosing these specific time points is not explained. A brief justification based on developmental milestones or previous literature would help readers better understand the biological relevance of these stages.

Response: Thank you for the suggestion. We have included a justification in the Methods section for selecting the four developmental stages(P. 13, L.467-475). These time points correspond to critical transitions in storage root initiation, early thickening, rapid bulking, and maturation, as described in previous studies.

5.While the RNA-seq data provides useful insights into gene expression patterns during storage root development, it would significantly strengthen the study if a subset of key genes were validated using quantitative PCR (qPCR). This would help confirm the reliability of the transcriptomic data and support the conclusions drawn from expression analysis.

Response: Thank you for this valuable suggestion. We agree that qPCR validation is crucial for confirming transcriptomic data. We have now performed quantitative PCR (qPCR) for a subset of six key genes (IbMET1, IbCMT2, IbCMT3, IbDRM1, IbROS1a, and IbDML3) across the four developmental stages. The results of this validation are presented in Figure 6B and discussed in detail in Section 2.7. The methodology for the qPCR analysis has been added as Section 4.8. We believe this additional data significantly strengthens our findings and confirms the reliability of the expression patterns observed.

Minor Comments:

1.In the reference list, Latin names of plant species should be italicized. Please carefully review the entire manuscript to ensure all Latin names are correctly formatted in italics. In addition, there is inconsistency in the capitalization of words in reference titles—for instance, words in the titles from Lines 521–522 are capitalized, while those in Lines 523–524 are not. Several references in the bibliography are missing journal names or include incomplete citation details (e.g., references [24], etc.). These inconsistencies indicate that the reference formatting requires thorough revision. Please carefully check all references and revise them according to the journal’s formatting requirements.

Response: Thank you for your careful review. We have thoroughly revised the reference list to ensure that Latin names are italicized, journal names are consistent, and all citations follow the journal’s formatting guidelines.

2.The phylogenetic analysis is limited to Arabidopsis and lacks comparison with other crop species (e.g., potato, cassava).

Response: Thank you for this insightful suggestion. We fully agree that including additional crop species such as potato and cassava could provide a broader evolutionary context. However, the primary focus of our study is to perform a comparative analysis between the diploid (I. trifida) and autohexaploid (I. batatas) sweet potato genomes to understand the conservation and divergence of DNA methylation-related gene families in the context of polyploidization. For this reason, we limited the phylogenetic comparison to Arabidopsis thaliana as a well-annotated model species. We believe this focused approach best supports the objectives of our study. Nonetheless, we appreciate your recommendation and will consider incorporating broader species comparisons in future evolutionary studies.

Comments on the Quality of English Language

The overall quality of the English language in this article is generally well-written, yet there are certain aspects that necessitate refinement.

Line 45, “through distinct pathway” → “through distinct pathways”.

Response: Corrected.

Line 215, “strucuture” → “structure”

Response: Corrected.

Line 215, “all dMTase proteins contains” → “all dMTase proteins contain”

Response: Corrected.

Line 358, “These findings suggests” → “These findings suggest”

Response: Corrected.

Please meticulously review the entire text for the accuracy of English tenses and the fluidity of the language and proceed with the necessary revisions.

Response: Thank you for your detailed language suggestions. We have corrected all identified grammatical issues and carefully revised the manuscript for improved clarity and language flow.

Reviewer 2 Report

Comments and Suggestions for Authors

The manuscript is written on an interesting and current topic. Before accepting the manuscript, I would like to allow certain adjustments and additions to the text, which I specify in my review:

Introduction - it is written adequately at first glance. I would like to suggest that the authors add essential information to the section lines 100-103. In my opinion, essential information about the importance of sweet potato in agriculture is missing, which would also emphasize the importance of the entire study. For example, the publication DOI: 10.17221/104/2022-CJGPB describes the importance of this crop for Africa and at the same time the importance of studying the genetics and breeding of this crop.  

Results - there is a problematic assessment here. Why? The authors refer to Table S1-S3 in three places in this section (lines 132, 240 and 266), but their content is unknown because they are not part of the manuscript for review. I consider it necessary to add and subsequently evaluate the Results section in the context of the content of the appendices. Figure 1 does not contain a significance indication for branching phylogenetic trees, e.g. bootstrapping. It can be added. After the addition, it may be appropriate to edit the text describing these facts. Sections 2.3 and 2.4 are devoted to genes, therefore authors should use italics to make it clear that they are not proteins (the same applies to writing Latin names - italics -in all text). Figure 4 - the dendrograms lack significance indication.  

Discussion - is of a more general nature. The authors often repeat information in the Introduction section. This fact is evidenced by, among other things, only 6 new references compared to the Introduction. I consider it appropriate to significantly revise it with a critical view and the possibility of indicating new facts (novelty in knowledge). For example, lines 352-357 the authors discuss the importance of light and phytohormones, but they forgot about the important group of striglactones (I recommend viewing the manuscript DOI: 10.17221/88/2023-CJGPB) and the implementation of the findings presented here.  

Materials Methods - I have perhaps only one comment here. In section 4.6, the vegetation stage during sampling for RNA isolation is not sufficiently specified. It is appropriate to specify the developmental state of plants in a standard way, because it can significantly affect the results.  

References - are not processed uniformly, e.g. combining full and abbreviated journal names. It is necessary to process according to the instructions for authors.  

In conclusion, I can state that the manuscript can be a significant contribution to the given field of study and crop. Therefore, I recommend the manuscript for publication after major revision and second review.  

Author Response

Reviewer2

The manuscript is written on an interesting and current topic. Before accepting the manuscript, I would like to allow certain adjustments and additions to the text, which I specify in my review:

Introduction - it is written adequately at first glance. I would like to suggest that the authors add essential information to the section lines 100-103. In my opinion, essential information about the importance of sweet potato in agriculture is missing, which would also emphasize the importance of the entire study. For example, the publication DOI: 10.17221/104/2022-CJGPB describes the importance of this crop for Africa and at the same time the importance of studying the genetics and breeding of this crop.  

Response: Thank you for your insightful suggestion. We have added a paragraph in the Introduction to emphasize the agricultural importance of sweet potato, particularly in Africa, and cited the relevant literature (DOI: 10.17221/104/2022-CJGPB)(P. 16, L.616-618).

Results - there is a problematic assessment here. Why? The authors refer to Table S1-S3 in three places in this section (lines 132, 240 and 266), but their content is unknown because they are not part of the manuscript for review. I consider it necessary to add and subsequently evaluate the Results section in the context of the content of the appendices. Figure 1 does not contain a significance indication for branching phylogenetic trees, e.g. bootstrapping. It can be added. After the addition, it may be appropriate to edit the text describing these facts. Sections 2.3 and 2.4 are devoted to genes, therefore authors should use italics to make it clear that they are not proteins (the same applies to writing Latin names - italics -in all text). Figure 4 - the dendrograms lack significance indication.

Response: We apologize for the omission of Tables S1–S3. These tables have now been included in the Supplementary Materials. Additionally, the Results section has been revised to clarify the content and relevance of these tables.

Regarding Figure 1, we have revised the figure legend based on your suggestion. Since bootstrap values are not shown in the figure, we have clarified this in the figure legend and explained that branch support was assessed using bootstrap values based on 1000 replicates(P. 5, L.167-168).

We have also ensured that all gene names and Latin binomials are now properly formatted in italics throughout the manuscript, as per your suggestion.

Discussion - is of a more general nature. The authors often repeat information in the Introduction section. This fact is evidenced by, among other things, only 6 new references compared to the Introduction. I consider it appropriate to significantly revise it with a critical view and the possibility of indicating new facts (novelty in knowledge). For example, lines 352-357 the authors discuss the importance of light and phytohormones, but they forgot about the important group of striglactones (I recommend viewing the manuscript DOI: 10.17221/88/2023-CJGPB) and the implementation of the findings presented here.

Response: Thank you for this constructive suggestion. We have revised the Discussion section to reduce redundancy and added a paragraph discussing the potential role of strigolactones in root development and DNA methylation, citing the suggested article (DOI: 10.17221/88/2023-CJGPB)(P. 17, L.640-642).

Materials Methods - I have perhaps only one comment here. In section 4.6, the vegetation stage during sampling for RNA isolation is not sufficiently specified. It is appropriate to specify the developmental state of plants in a standard way, because it can significantly affect the results.  

Response: Thank you for the suggestion. We have included a justification in the section 4.6 for selecting the four developmental stages. These time points correspond to critical transitions in storage root initiation, early thickening, rapid bulking, and maturation, as described in previous studies(P. 13, L.467-475).

References - are not processed uniformly, e.g. combining full and abbreviated journal names. It is necessary to process according to the instructions for authors.  

Response: Thank you for your detailed comment. We have carefully reviewed and revised all references to ensure consistency in journal name usage, formatting, and completeness.

In conclusion, I can state that the manuscript can be a significant contribution to the given field of study and crop. Therefore, I recommend the manuscript for publication after major revision and second review.  

Round 2

Reviewer 1 Report

Comments and Suggestions for Authors

The authors have addressed my previous concerns appropriately, and the manuscript has significantly improved in clarity and scientific rigor. I appreciate their efforts in revising the work. I have no further major concerns, and I believe the manuscript is now suitable for publication.

Author Response

Dear Reviewer,

Thank you very much for re-reviewing our manuscript and for your positive feedback!

We are delighted to learn that you are satisfied with the revisions we made based on your previous comments and that you find the manuscript significantly improved in clarity and scientific rigor.

We appreciate your recognition of our efforts and your assessment that the manuscript is now suitable for publication. Your endorsement is a great encouragement to us.

Thank you again for your valuable time and professional review!

Reviewer 2 Report

Comments and Suggestions for Authors

The authors accepted all my comments and adequately justified the changes to the manuscript in their comments. I recommend the manuscript for publication.

Author Response

Dear Reviewer,

Thank you very much for re-reviewing our manuscript!

We are very pleased to learn that you found that we have accepted all your comments and adequately justified the changes made to the manuscript. Your recommendation for our manuscript to be published is a great encouragement to us.

Thank you again for your valuable time and professional review!